# Technical Note: Novel triple O₂ sensor aquatic eddy covariance instrument with improved time shift correction reveals central role of microphytobenthos for carbon cycling in coral reef sands

Alireza Merikhi[1], Peter Berg[2], Markus Huettel[1]

[1] Department of Earth, Ocean and Atmospheric Science, Florida State University, Tallahassee, FL 32306-4520, USA
[2] Department of Environmental Sciences, University of Virginia, Charlottesville, VA 22904-4123, USA

*Correspondence to*: Markus Huettel (mhuettel@fsu.edu)

**Abstract.** The aquatic eddy covariance technique stands out as a powerful method for benthic O₂ flux measurements in shelf
environments because it integrates effects of naturally varying drivers of the flux such as current flow and light. In conventional eddy covariance instruments, the time shift caused by spatial separation of the measuring locations of flow and O₂ concentration can produce substantial flux errors that are difficult to correct. We here introduce a triple O₂ sensor-eddy covariance instrument (3OEC) that by instrument design eliminates these errors. This is achieved by positioning three O₂ sensors around the flow measuring volume, which allows calculating the O₂ concentration at the point of the current flow
measurements. The new instrument was tested in an energetic coastal environment with highly permeable coral reef sands colonized by microphytobenthos. Parallel deployments of the 3OEC and a conventional eddy covariance system (2OEC) demonstrate that the new instrument produces more consistent fluxes with lower error margin. 3OEC fluxes in general were lower than 2OEC fluxes, and the nighttime fluxes recorded by the two instruments were statistically different. We attribute this to the elimination of uncertainties associated with the time shift correction. The deployments at ~10 m water depth
revealed high day- and nighttime O₂ fluxes despite relatively low organic content of coarse sediment and overlying water. High light utilization efficiency of the microphytobenthos and bottom currents increasing pore water exchange facilitated the high benthic production and coupled respiration. 3OEC measurements after sunset documented a gradual transfer of negative flux signals from the small turbulence generated at the sediment water interface to the larger wave-dominated eddies of the overlying water column that still carried a positive flux signal, suggesting concurrent fluxes in opposite directions depending
on eddy size and a memory effect of large eddies. The results demonstrate that the 3OEC can improve the precision of benthic flux measurements, including measurements in environments considered challenging for the eddy covariance technique, and thereby produce novel insights into the mechanisms that control flux. We consider the fluxes produced by this instrument for the permeable reef sands the most realistic achievable with present day technology.

## 1 Introduction

This study introduces a new eddy covariance instrument and demonstrates its functionality through a measuring series addressing benthic oxygen flux in a dynamic backreef area covered by highly permeable carbonate sands. The aquatic eddy

covariance technique is a powerful technique for quantifying fluxes at the seafloor as it measures over any type of substrate, and integrates over a relatively large area (Berg et al., 2003;Lorrai et al., 2010;McGinnis et al., 2008). This non-invasive technique derives flux by averaging the turbulent vertical advective transport of $O_2$ (or other solute) above the sediment over time (Berg et al., 2003; Lorrai et al., 2010; Mcginnis et al., 2008b). Since the vast majority of the recorded flux signals originate from the seafloor upstream of the instrument (termed "footprint"), the flux measurements include the effects of the natural flow and light fields, and the benthic sedimentary community (Berg et al., 2003; Lorrai et al., 2010; Mcginnis et al., 2008b; Huettel et al., 2020). The eddy covariance method thus can produce flux data in dynamic shelf ecosystems such as coral reefs that are strongly influenced by flow and light (Long et al., 2013; Long, 2021; Huettel et al., 2020). The technology so far has been adapted to measure temperature, salinity, oxygen, hydrogen, sulphide, and nitrate fluxes (Mcginnis et al., 2011; Johnson et al., 2011; Long et al., 2015; Crusius et al., 2008; Weck and Lorke, 2017).

Since the fluxes are calculated from minute concentration changes measured at a high frequency required to account for all water movements that transport the solute, it is critical that the flow data are accurately aligned in time with the associated solute data. This presently is a potential source of error as conventional aquatic eddy covariance instruments cannot measure current velocities and solute concentrations in the same location. Typically, current velocities are recorded with an acoustic Doppler velocimeter (ADV), and solutes with a fast-responding electrochemical or optical sensor (Kuwae et al., 2006; Berg et al., 2003; Reimers et al., 2012; Attard et al., 2016; Donis et al., 2016; Glud et al., 2010; Mcginnis et al., 2014; Lorke et al., 2013; Huettel et al., 2020). The tip of the solute sensor in these instruments is positioned at a few centimetres horizontal distance from the ADV's measuring volume to prevent disturbances of the flow and the acoustic ADV signal. This spatial separation of flow and solute measurements causes a misalignment between the two measurement time series, which requires a time shift correction of the data. In environments with dynamic currents, this misalignment changes continuously as direction and velocity of the turbulent flow varies. Algorithms were developed that shift the $O_2$ data in time such that they are synchronized with the velocity data (Mcginnis et al., 2008a; Berg et al., 2015; Reimers et al., 2016). A common procedure is to move a short sequence (e.g., 15 min) of solute data in time relative to the current flow data recorded at that time until a maximum in flux is reached. In steady unidirectional flow, this procedure largely can eliminate time shift errors, but it is difficult to apply an effective correction in dynamic settings (Donis et al., 2015; Reimers et al., 2016). Since the rapid changes in solute concentration and vertical flow velocity are relatively small and affected by signal noise, a distinct maximum in flux may not be found when time shifting the data, which can result in erroneous corrections and fluxes. Furthermore, wave orbital motion in shelf environments produces oscillating bottom currents that may change in magnitude and direction at the time scale of seconds, complicating a correct alignment of the data and producing further potential sources of uncertainty in the flux calculations. In the conventional eddy covariance instruments with one or two solute sensors, the cumulative effect of small errors in the time shift correction thus can lead to significant under- or overestimates of the flux, which in extreme cases can reverse the direction of the calculated flux relative to the true flux (Berg et al., 2015;

Reimers et al., 2016). To remove this potential source of error, we designed a triple $O_2$ sensor eddy covariance instrument (3OEC) that eliminates the uncertainties caused by the spatial separation of flow and concentration measurements.

The instrument was tested in the Florida Keys at an exposed inner shelf site with carbonate sands, clear oligotrophic water and substantial wave action, i.e., in an environment considered challenging for eddy covariance measurements due to the low
particle concentrations in the water (ADV measurements rely on sound reflection from particles) and the dynamic flows (causing the data misalignments addressed in this study). We selected that site because carbonate sand beds are an integral part of reef environments and may play a central role in their carbon and nutrients cycles (Eyre et al., 2018; Santos et al., 2011; Cyronak et al., 2013). As warm water coral reefs grow in shallow, high-energy environments, these sediments typically are dominated by highly permeable coarse sands (Yahel et al., 2002; Harris et al., 2015) colonized by
microphytobenthos (Werner et al., 2008; Jantzen et al., 2013). Owing to the rapid pore water exchange facilitated by the high permeability, biogeochemical processes in the sediment surface layers can respond almost instantly to changes in flow, organic matter input, and light (Huettel et al., 2014). This makes the eddy covariance technique the unrivalled method for measuring interfacial fluxes in this environment provided that potential errors associated with time shift correction in dynamic flows can be removed.

The goals of this study therefore were to develop an eddy covariance instrument not affected by uncertainties caused by spatial separation of flow and solute measuring points and to demonstrate its functionality through measurements in a dynamic coral reef environment. The instrument measured $O_2$ flux, which is considered a good proxy for benthic production and respiration (Berg et al., 2013; Glud, 2008). Comparison with parallel measurements with a conventional eddy covariance
instrument reveals the improvements achieved by the new instrument design.

## 2 Methods

### 2.1 Triple $O_2$ sensor eddy covariance instrument (3OEC)

The new 3OEC instrument utilizes a data averaging approach to remove misalignments in time caused by the spatial separation of flow and $O_2$ measurements and thereby eliminates potential errors caused by time shift corrections. The 3OEC
measures simultaneously with three $O_2$ fibre optodes positioned at 120 degrees angular spacing in the same horizontal plane around the centre point of the water volume where current flow is measured by the ADV (Fig. 1). Assuming approximately linear concentration gradients within the 6.4 cm distance between the $O_2$ sensors - justifiable according to planar optode readings of water column oxygen distributions in turbulent flows (Glud et al., 2001; Larsen et al., 2011; Oguri et al., 2007) - the $O_2$ concentration at the location where the flow is measured can be calculated through averaging of the three
simultaneous sensor signals. The three optode tips, positioned at the corners of an equilateral triangle, present three equidistant points on a circle with the measuring volume of the ADV at its centre. For linear concentration gradients between

the three measuring points, it can be proven with the law of sines that the $O_2$ concentration at the centre of this circle corresponds to the average of the three sensor signals (Fig. 1e).

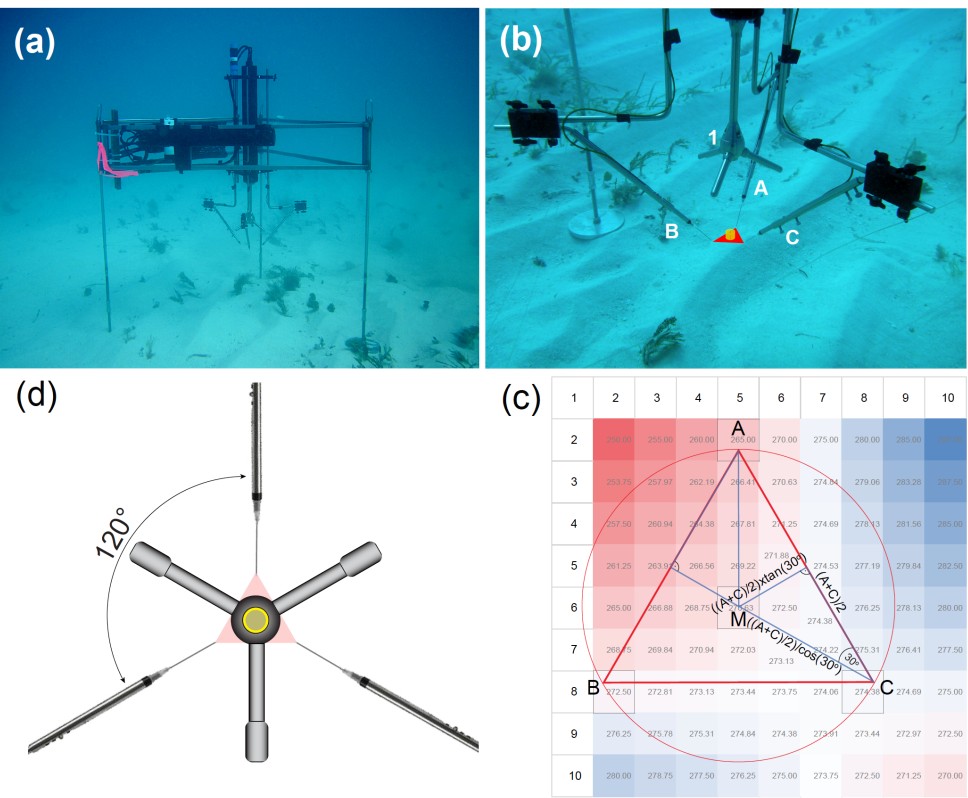

**Figure 1: The triple $O_2$ sensor eddy covariance instrument (3OEC). (a) The 3OEC deployed over carbonate sand at the study site in the Florida Keys. (b) Positioning of the ADV sensor head (1) and the three $O_2$ sensors (A, B, C) (2). The average $O_2$ concentration calculated from the readings of the three $O_2$ sensors approximates the concentration at the geometric centre of the red triangle (side lengths 6.4 cm) defined by the three $O_2$ sensor tips. This geometric centre is also the centre of the flow measuring volume of the ADV, marked as yellow cylinder. (c) Vertical view of the positioning of the $O_2$ sensors around the ADV measuring volume (yellow circle) that is located 15 cm below the central sensor stem. The sensor tips are located at 3.0 cm horizontal distance from the edge of the ADV measuring volume. The red triangle is the vertical view of the red triangle shown in (b). (d) Proof of the equivalence of the average of the three sensor readings and the concentration at the centre of the equilateral triangle defined by the positions of the three optode sensor tips (red triangle in (b) and (c)). We postulate that the concentration gradients between the sensor tips are linear (see also text). Accordingly, the half-way point of a side of the triangle corresponds to the average concentration measured by the optodes at the two endpoints of that side. Applying the law of sines and triangle congruence criteria (transitive property of congruence, angle bisector theorem, converse of angle bisector theorem), the concentration at the centre point (M) of the equilateral triangle (ABC) equals the concentration at the vertex (M) of the right triangle defined by one vertex of the equilateral triangle (C in above example) and the midpoint ((A+C)/2 in above example) between that vertex and the second vertex (A in above example) on that line. According to the congruence criteria, this is valid for analogous, congruent right triangles constructed on the other sides of the triangle. As these triangles are based on the average concentration of two vertices of the equilateral triangle, it follows that the centrepoint concentration is equivalent to the average concentrations measured by the three sensors. The heat map visualizes the equivalence of the concentrations at the centre of the equilateral triangle calculated using this approach and the average of the three sensor signals. (Photographs: Huettel)**

The optodes are ultra-highspeed $O_2$-needle sensors (Pyroscience™- OXR430-UHS, Table A1) with a response time of 200-300 ms (Merikhi et al., 2018). The three optodes are pointing downward at a 45° angle, with their sensing tips positioned at 3.7 cm horizontal distance from the centre of the flow measuring volume. This placement, within the recommended distance of 10 Kolmogorov scale lengths from the ADV measuring volume (Lorrai et al., 2010), prevents any disturbance of the flow in that volume and potential interferences with the acoustic pulses of the ADV. The sensors are read by three FireStingO2-Mini $O_2$-meters (Pyroscience™, Table A2). The ADV is a Nortek™ Vector acoustic Doppler velocimeter (Table A3) that measures the 3D velocity field within a cylindrical measuring volume (1.5 cm diam. x 1.5 cm, located 15 cm below the central acoustic transducer) at a sampling rate of 32 Hz. A DataQ DI-710-UH USB data logger (14-bit A/D conversion) records simultaneously the output of $O_2$-meters and the ADV at a rate of 64 Hz to prevent aliasing. $O_2$-meters and data logger are contained in an underwater housing (A.G.O Environmental Electronics) fitted with three Pyroscience™ fibre feed through plugs for connecting the $O_2$ sensors, and Impulse micro inline plugs for connecting the ADV and external battery (4 x Lithium-Ion 12 V, 50 Wh). The ADV, $O_2$ meter housing, and battery pack are mounted on a stainless-steel tripod with 1.2 m side length and 1.2 m height (Berg and Huettel, 2008). In addition, the frame carries a MiniDot $O_2$-logger (PME) and an Odyssey PAR-logger (Dataflow Systems) for collection of temperature and $O_2$ reference data (once per minute) and photosynthetically active radiation (PAR) data (once per 10 minutes), respectively.

## 2.2 Dual $O_2$ sensor eddy instrument (2OEC)

An eddy covariance instrument with conventional sensor configuration (2OEC) was deployed parallel to the 3OEC to analyse the potential flux error caused by the time shift and the effectiveness of standard data corrections. The 2OEC, described in detail in Huettel et al. (2020), measures simultaneously with two $O_2$-optodes positioned on one side of the ADV measuring volume with their measuring tips 1 cm horizontally apart. Deployments of this instrument at the same study site in the Florida Keys simultaneously with benthic advection chambers (Huettel and Gust, 1992; Janssen et al., 2005; Huettel et al., 2020) produced eddy covariance fluxes ($3.7 \pm 0.9$ mmol m$^{-2}$ h$^{-1}$) that were similar to those of the chamber fluxes ($3.9 \pm 3.0$ mmol m$^{-2}$ h$^{-1}$) during daytime, and of similar order of magnitude during nighttime (2OEC: $-2.5 \pm 1.3$ mmol m$^{-2}$ h$^{-1}$, chambers: $-3.4 \pm 0.8$ mmol m$^{-2}$ h$^{-1}$). These fluxes obtained with an independent measuring technique corroborate the magnitude of the eddy covariance fluxes, but it should be noted that the chambers do not account for changes in flow and organic matter supply during the incubation, which both can have significant influence on the flux. While the chambers under relatively steady conditions and short incubation periods can produce fluxes similar to those recorded by eddy covariance instruments, discrepancies between fluxes measured by the two techniques were observed in dynamic environments (Berg et al., 2013).

## 2.3 Data processing

Eddy covariance flux calculations are based on the assumption that the flux signal is transported by a bottom current with steady state mean flow and $O_2$ concentration that reaches the instrument unobstructed after passing the footprint area (Massman and Lee, 2002; Baldocchi, 2003; Kuwae et al., 2006; Berg et al., 2007). In coastal environments, such conditions rarely are met, requiring post processing of the flux data to correct for infringements of these assumptions as well as errors caused by technical limitations (Holtappels et al., 2013; Reimers et al., 2016; Huettel et al., 2020). We applied the same routine corrections to 3OEC and 2OEC data for compensation of design and sensor limitations as well as for non-steady-state $O_2$ concentrations in the water column. The unfiltered flow and $O_2$ data records were reduced from 64 Hz to 8 Hz by averaging, which reduced noise but maintained sufficient resolution to describe the entire frequency spectrum carrying the flux signal. For each 8 Hz time point, the average signal of the three $O_2$ sensors of the 3OEC was calculated to determine an estimate of the $O_2$ concentration in the centre of the ADV measuring volume. Similarly, the signals of the two sensors of the 2OEC were averaged to produce mean concentrations. $O_2$ fluxes then were calculated based on these averages as well as based on the signals of each individual sensor using the software EddyFlux 3.2 (Berg unpublished). The software determines mean $O_2$ base concentrations for 15 min time segments through Reynolds decomposition (Lorrai et al., 2010; Berg et al., 2009; Lee et al., 2004). Within each 15-minute interval, the mean $O_2$ concentration $\overline{O_2}$ (defined as a least-square linear fit to the data) then is subtracted from each 8 Hz $O_2$ data point to arrive at the instantaneous $O_2$ fluctuation $O_2'$ for that time point. The instantaneous vertical velocity $V_z$' is determined using the same procedure. The flux at each 8 Hz time point is calculated by multiplying the instantaneous vertical velocity and associated instantaneous $O_2$ concentration. The changes in fluxes were added over time to produce cumulative flux curves. For three consecutive time intervals (42 to 95 minutes in length) with undisturbed flux during day and night, slopes of these curves then were calculated to determine light- and dark fluxes, respectively. In the following text, fluxes based on the averaged signal of 3 (3OEC) or 2 (2OEC) $O_2$ sensors were termed "3S-flux" and "2S-flux" respectively. Single sensor fluxes were termed "1S-flux" and uncorrected fluxes "raw" fluxes.

Standard corrections, abbreviated in this text by single letters, were applied to the flux data to reduce potential errors caused by instrument tilt (R), wave effects (W), time shift (T) - caused by spatial separation of sensors (2OEC) and sensor response time - , and changes in water column $O_2$ storage (S) (Berg et al., 2015; Mcginnis et al., 2008a; Lorke et al., 2013; Huettel et al., 2020). Influence of potential instrument tilt (R) on flux was tested and corrected when necessary, through rotation of the velocity data so that the mean transverse and vertical velocity was nullified (Lee et al., 2004; Lorke et al., 2013; Lorrai et al., 2010). Similarly, wave rotation (W) was rectified by rotating the flow velocity field so that SD($V_y$) and SD($V_z$) reached a minimum (SD represents 1 standard deviation)(Berg et al., 2015; Berg et al., 2013). Time shifts (T) were rectified through applying time shift corrections to the $O_2$ data that produced the maximum absolute fluxes (Mcginnis et al., 2008a; Berg et al., 2015; Berg et al., 2003; Reimers et al., 2016; Fan et al., 1990). Effects of large-scale variations in the average water column

$O_2$ concentration (S) were compensated through applying an $O_2$ storage term ($J_{St} = \int_0^h dC/dt\ h$, with $dC/dt$ : change of the average $O_2$ concentration over time, calculated through linear detrending of the measured $O_2$ data over 15 minute intervals, h = height of the measuring volume) (Holtappels et al., 2013; Rheuban et al., 2014). Furthermore, acceleration or deceleration of current flows can alter the $O_2$ concentration profile and thereby temporarily modulate vertical flux (Holtappels et al., 2013). Our data analysis indicated that the temporal flux variations caused by transient velocity changes largely cancelled out over time, and a correction for transient velocity changes was not applied.

Daytime was defined as the period between sunrise and sunset. To determine the significance of differences in fluxes measured by the two instruments, the paired t-test was utilized. Error margins are reported as $\pm\ 1$ standard deviation unless stated otherwise.

**2.4 Instrument deployments**

The 3OEC and 2OEC were deployed at $9 \pm 1$ m water depth on an exposed backreef carbonate platform in the Florida Keys (24° 43.523'N, 80° 49.855'W, Fig. 1a) on July 11, 13, 15, and 16, 2017. Prior to the deployments, the instruments were synchronized in time. SCUBA-divers placed the two instruments 10 m apart, along a transect perpendicular to the main southwest-northeast flow direction. The tripods were rotated such that the X-axis of the ADVs was aligned with the main current direction, and the measuring volumes of the ADVs were adjusted to 35 cm above the average sediment surface level. The seafloor here is covered by highly permeable medium carbonate sand (median grain size: 440 µm, permeability: $3.2 \times 10^{-11} \pm 1.2 \times 10^{-12}\ m^2$) with relatively low carbon content (0.23% ± 0.05% sed. dw.) and colonized by microphytobenthos (Chlorophyll a: $4.9 \pm 0.1\ \mu g\ g^{-1}$ sed. dw.). During the deployment week, water temperatures averaged $29.9 \pm 0.3$ °C and salinity $35.0 \pm 0.5$. Bottom current velocities ranged from 5 to 14 cm $s^{-1}$. Waves increased from 11 July to 15 of July, when maximum wave heights of 90 cm were reached, and then dropped again on 16 July. The weather was mostly sunny with some scattered clouds resulting in relatively high light intensities at the seafloor reaching 392 µmol photons $m^{-2}\ s^{-1}$. On each measuring day, the instruments were deployed during daylight time to include the effect of benthic photosynthesis and were retrieved the following day for data download.

**3 Results**

**3.1 Benthic fluxes**

The $O_2$ fluxes measured by the 3OEC were lower and less variable than those recorded by the 2OEC (Fig. 2ab). Daytime 3OEC $O_2$ fluxes averaged $5.2 \pm 0.6$(SE) mmol $m^{-2}\ h^{-1}$, nighttime fluxes $-2.8 \pm 0.6$(SE) mmol $m^{-2}\ h^{-1}$, characterizing the permeable carbonate sand bed as a site of high carbon turnover and net autotrophy in July 2017. Average 3OEC daytime fluxes were 7% lower and nighttime fluxes 38% lower than the respective 2OEC fluxes (day: $5.6 \pm 0.8$(SE) night: $-3.9 \pm 0.5$(SE) mmol $m^{-2}\ h^{-1}$). The difference in the nighttime fluxes between the two instruments was statistically significant (p=

0.04685, p(x≤T) = 0.02342, T =-3.268, DF=3), while the difference in daytime fluxes was not (p=0.08077, p(x≤T) = 0.9596, T = 2.5944, DF=3). The trajectories of the cumulative fluxes were similar in both instruments, but in the 3OEC, the additional sensor and elimination of errors associated with time shift corrections reduced fluctuations of the averaged signal trajectories (Fig. 2c).

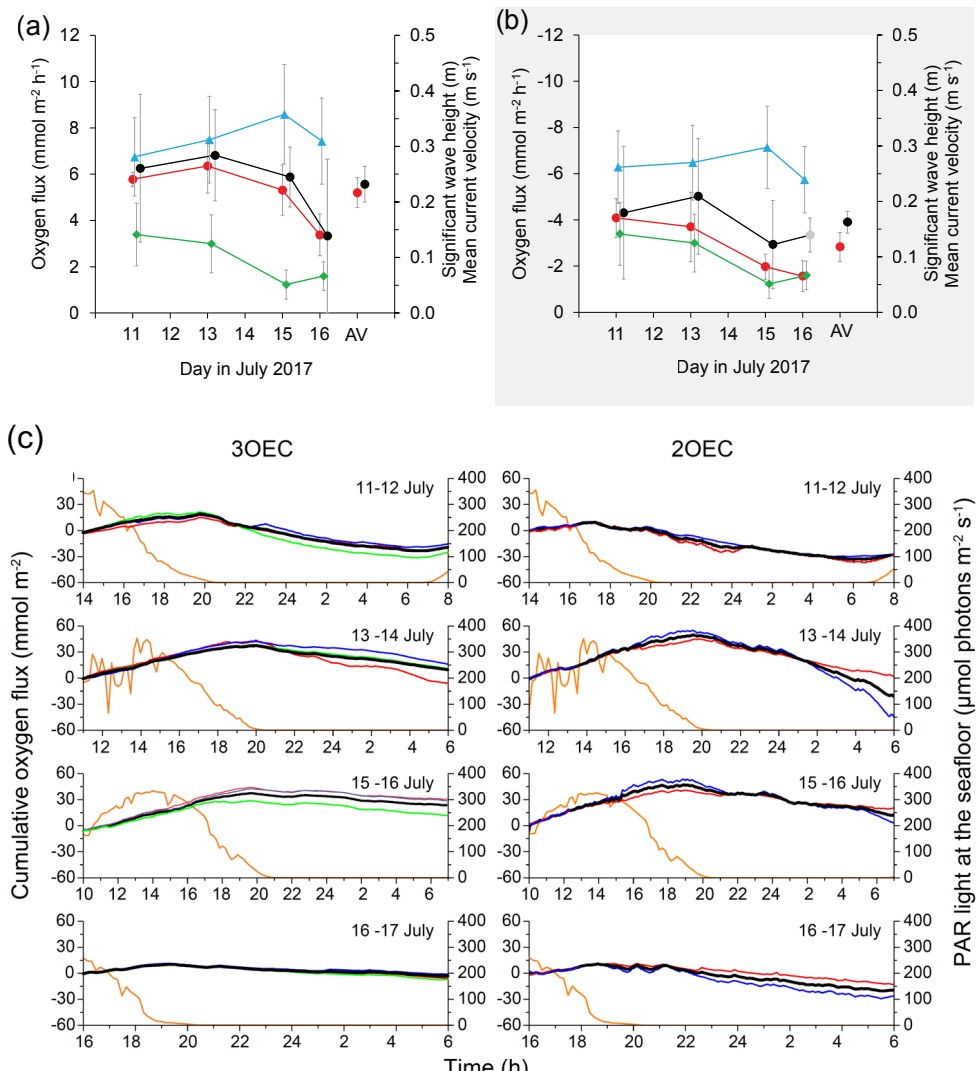

**Figure 2: Fluxes and cumulative flux trajectories recorded during the deployment week. (a) Day time and (b) nighttime O₂ fluxes measured by the 3OEC (red circles and line) and 2OEC (black circles and line), and significant wave heights (light blue triangles and line) and current flows (green diamonds and line) recorded in July 2017. Note reversed Y axis scale for (b) night fluxes. Grey circle in (b) indicates data point compromised by sensor deterioration. Error bars depict standard deviation, except for the 11-16 July averages (single circles on right side of panes), where error bars present standard error. (c) Comparison of cumulative fluxes measured by the 3OEC (left column) and 2OEC (right column) on July 11-12, 13-14, 15-16 and 16-17 (top to bottom) with the respective light (PAR) intensities (orange lines) at the seafloor. Sensor 1: red, Sensor 2: blue, Sensor 3: green, average sensor signals: thick black lines.**

In the 3OEC, elimination of the time shift caused by spatial separation of $O_2$ and flow measurements does not completely remove time shift errors from the raw fluxes. The remaining time shift errors are caused by the response time of the $O_2$ sensors (0.2-0.3 s) and temporary distortions of the $O_2$ concentration field (Fig. 3).

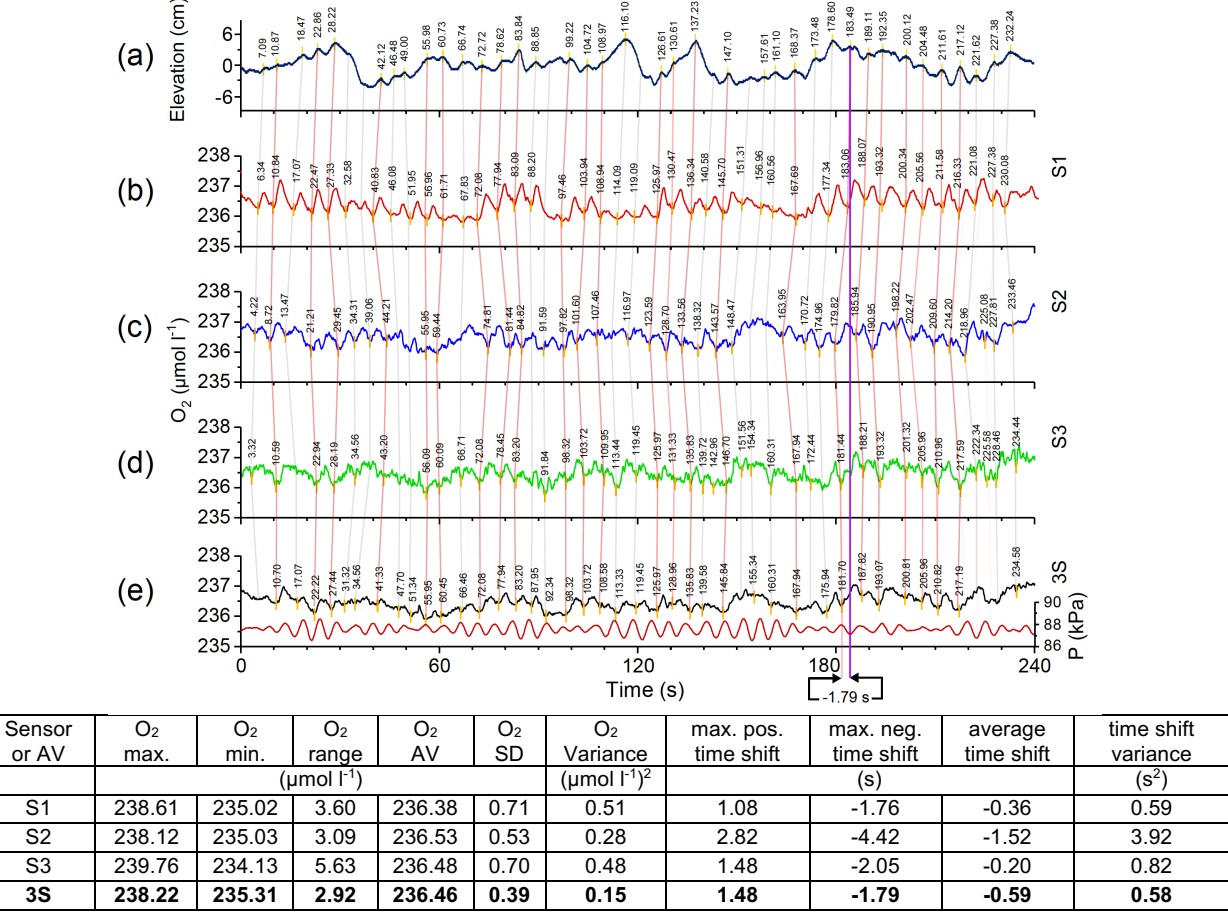

| Sensor or AV | $O_2$ max. | $O_2$ min. | $O_2$ range | $O_2$ AV | $O_2$ SD | $O_2$ Variance | max. pos. time shift | max. neg. time shift | average time shift | time shift variance |
|---|---|---|---|---|---|---|---|---|---|---|
| | ($\mu$mol l$^{-1}$) | | | | | ($\mu$mol l$^{-1}$)$^2$ | (s) | | | (s$^2$) |
| S1 | 238.61 | 235.02 | 3.60 | 236.38 | 0.71 | 0.51 | 1.08 | -1.76 | -0.36 | 0.59 |
| S2 | 238.12 | 235.03 | 3.09 | 236.53 | 0.53 | 0.28 | 2.82 | -4.42 | -1.52 | 3.92 |
| S3 | 239.76 | 234.13 | 5.63 | 236.48 | 0.70 | 0.48 | 1.48 | -2.05 | -0.20 | 0.82 |
| **3S** | **238.22** | **235.31** | **2.92** | **236.46** | **0.39** | **0.15** | **1.48** | **-1.79** | **-0.59** | **0.58** |

**Figure 3. Four-minute interval of nighttime data recorded on July 13th (4:53:20-4:57:20) comparing the simultaneous elevation, oxygen and pressure readings recorded by the 3OEC. Elevation expresses the instantaneous relative elevation of a water parcel that is moved up and down in the water column (see text). (a) elevation (blue line) and (b, c, d) the associated $O_2$ concentrations recorded by the three optodes (red, blue, and green lines) and their (e) average (black line). Brown line in (e) depicts pressure P (scale right Y axis) at the height of the ADV. The data were smoothed by a 2.5 s running average. In absence of a time shift, a minimum in $O_2$ change occurs when water displacement is zero, and points with zero $O_2$ change therefore are connected in this graph to points with no elevation change. Since this is a nighttime recording, $O_2$ minima cross-correlate with elevation maxima. Vertical red lines connect elevation maxima and associated $O_2$ minima as identified by the Origin lab 2017 software peak-finding algorithm (analysis of 2nd derivative). Grey vertical lines do the same but in these cases one of the minima or maxima could not be identified by the peak-finding algorithm (manual fit). The instance of the largest temporary time shift (-1.79 s) between elevation maximum and the corresponding 3S-$O_2$-minimum observed within this 4 min interval is indicated by the vertical purple line. The data listed below the graph reveal how the averaging of the $O_2$ signals reduces the variance of the $O_2$ signal in the flow measuring volume relative to the individual $O_2$ signals.**

Within the oxygen gradient near the seafloor, vertical water movement associated with wave orbital motion causes $O_2$-oscillations at a fixed point above the sediment, i.e., at the ADV's flow measuring point. During nighttime, $O_2$ signal minima occur at the maxima of water parcel elevation, as water originating near the $O_2$-consuming sediment surface is moved up within the water column. Elevation $z$ expresses the instantaneous relative elevation of a water parcel that is moved up and down at the velocity $V_z$ and can be estimated as $z = \int V_z \, dt$ (Berg et al., 2015). In the 4-minute recording example shown in

Figure 3, nearly parallel vertical connecting lines between the $O_2$ concentration minima recorded by the three optodes during time intervals with reduced wave activity (e.g., 40-60 s, 90-120 s, 210-220 s) confirm that the sensor response times were similar and consistent. Bending in the connecting lines during periods with increased wave activity (reflected by larger pressure oscillations, Fig. 3e) reveals distortions of the $O_2$ concentration field at the scale of the oxygen sensor spacing (i.e., within the triangle in Fig. 1b). Since the observed time shifts between elevation maxima and the associated $O_2$ signal minima

are positive as well as negative, the bending cannot be attributed to optode response characteristics, implying that it is caused by distortions in the $O_2$ concentration field. Such distortions temporarily move the 3S signal slightly off centre in the flow measuring volume, producing varying time shifts between 3S signal and velocity data. In this example, a maximum time shift of 1.79 s briefly was reached at $t = 183.49$ s, lasting less than 6 s. The total time shift caused by sensor response time plus transitory shifts produced by distortion during these 4 minutes averaged -0.36, -1.52 and -0.20 s for the three sensors,

respectively, and -0.59 s for the 3S signal. To compensate for sensor response time, a time shift correction was included in the corrections (STW) used when calculating all 3S fluxes. The temporary time shifts caused by transitory concentration field distortions average out over time as reflected in the 3S variances that were 1.9 to 3.4-times smaller than the 1S variances, and a correction for concentration field distortion was not applied.

3OEC fluxes based on the 3S signal had lower standard errors than fluxes based on individual 1S signals, i.e., standard error was reduced by 27% ± 35%(1SD) during daytime and 114% ± 158%(1SD) during nighttime (Fig. 4a). The normalized daytime 1S-fluxes deviated 3% (3OEC) and 2% (2OEC), nighttime fluxes 18% (3OEC) and 26% (2OEC) from the respective normalized 3S- and 2S-fluxes (Fig. 4b). Corrections for storage (S) and time shift (T, in 3OEC to correct for response time) increased raw flux, while corrections for wave rotation (W) and instrument tilt (R) reduced it (Fig. 4c).

Applying a combination of storage, time shift and wave rotation corrections (STW) led to the best agreement between 1S-fluxes as well as to the strongest enhancement of the raw flux (Fig. 4c,d), as previously found in 2OEC deployments conducted at the same study site (Huettel et al., 2020).

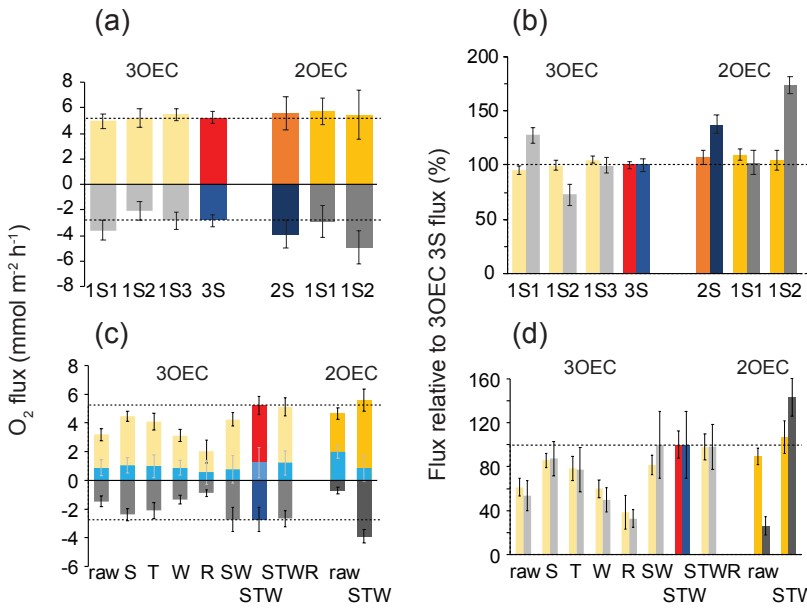

**Figure 4: Comparison of fluxes based on single sensor signals and average signals, and the effects of flux corrections.** Error bars represent standard error. (a) Comparison of the daytime (yellow/orange) and nighttime (grey/dark grey) fluxes averaged over all days based on individual sensors and sensor averages (day: red/brown, night: blue/darkblue). (b) normalized 1S, 2S and 3S fluxes, color code as in (a). The 3S flux (red/blue) was set to 100% (all data in (a) and (b) are STW- corrected). (c) Effect of corrections on 3S and 2S daytime, nighttime and 24h fluxes (light blue) averaged over all days (error bars SE). raw: not corrected, S: Storage-corrected, T: Time shift-corrected, W: Wave rotation-corrected, R: Rotation-corrected. SW, STW, STWR are combinations of the above corrections. (d) normalized differences between the 3OEC STW-corrected average fluxes (set to 100%) and fluxes with no correction or different corrections recorded with the 3OEC and the 2OEC. Column color coding in (c) and (d) as listed for (a). Dotted lines allow comparison of the fluxes with the 3S fluxes. Corresponding graphs based on the individual sensor readings are included in the supplemental materials.


At our study site, waves were relatively high for this shallow environment (wave height up to 10% of water depth), and wave orbital motion influenced water movement and pressure near the seafloor during the entire study (e.g., Fig. 3e). Yet, fluxes scaled with the average unidirectional bottom current velocity, which slowed during the deployment week ($\sim$30 mmol m$^{-2}$ h$^{-1}$ flux increase or decrease per m s$^{-1}$ flow decrease, Fig. 5a), and not with significant wave height ($R^2<0.04$, Fig. 2ab) that

increased during the study except the last day. On that last day, significant wave heights were nearly identical to those recorded three days earlier (Fig. 2ab, day: 0.31, 0.31 m, night: 0.27, 0.24 m, for 13-14 and 16-17 July, respectively), but daytime fluxes decreased by 47% and sand nighttime fluxes by 58% between these deployments. Light was ruled out as cause for the decreases in the fluxes over time because light conditions were similar between deployment days (858 ±165 mmol photon m$^{-2}$ surface PAR for the overlapping time period 17:00-20:00). Improved precision and the generally lower

fluxes in the 3OEC were reflected in the community photosynthesis-irradiance (PI) curves (Bernardi et al., 2015). The 3OEC predicted a slightly lower maximum gross benthic primary production (GPP) of 9.9 mmol O$_2$ m$^{-2}$ h$^{-1}$ ($R^2$: 0.999) than the

2OEC (10.7 mmol $O_2$ m$^{-2}$ h$^{-1}$, $R^2$: 0.998, Fig. 5b) as well as a lower light utilization efficiency (LUE, ratio between GPP and PAR, 3OEC LUE 12.3% lower than 2OEC LUE at 10 µmol photon m$^{-2}$ s$^{-1}$ and 7.4% lower at 350 µmol photon m$^{-2}$ s$^{-1}$ (Fig. 5c)). Due to the scatter in the data, these differences in GPP maxima and LUE were statistically not significant.


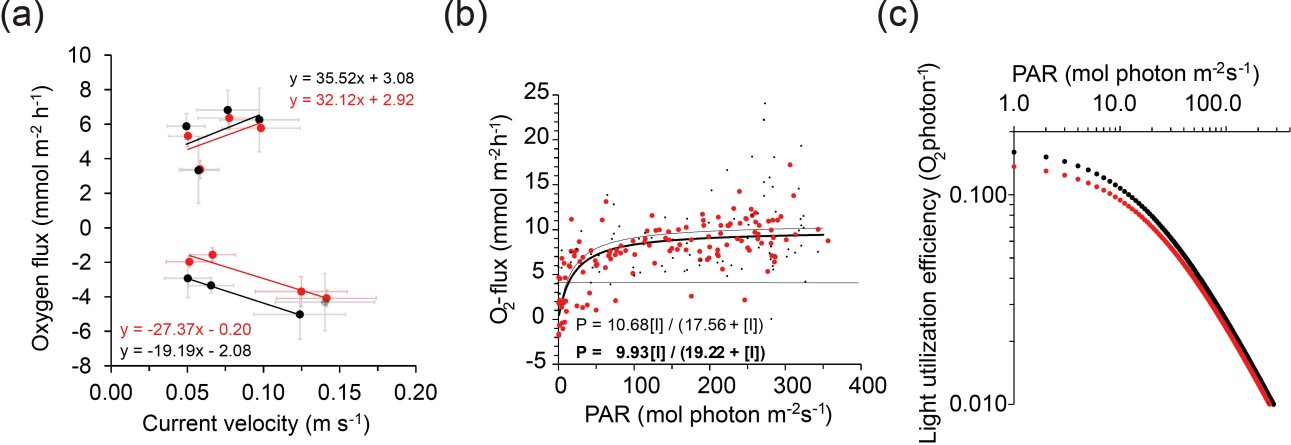

**Figure 5: Effect of flow and light on flux. (a) Effect of flow velocity on day- and nighttime fluxes measured with the 3OEC (red circles and line) and 2OEC 4 (black circles and line). The data points indicate the average fluxes calculated for light daytime and dark nighttime periods, separated at 20:00, plotted against the average flow velocity for the respective time periods. Compromised data point from the 16 July 2OEC deployment excluded from regression (grey circle). Error bars represent standard error. (b) Increase of daytime fluxes with increasing light intensity at the seafloor. Black curves depict photosynthesis-irradiance-curves (red circles, thick line: 3OEC, black dots, thin line: 2OEC) calculated using Michaelis-Menten kinetics (P = $P_{max}$ [I] / ($K_I$ + [I]), with P: photosynthetic rate at a given light intensity, $P_{max}$: maximum potential photosynthetic rate, [I]: light intensity, $K_I$: half-saturation constant, i.e., the light intensity at which the photosynthetic rate proceeds at ½ $P_{max}$). Horizontal black line indicates approximate level of daytime respiration. (c) Light utilization efficiency of the benthic community based on data shown in (b) (red circles: 3OEC, black circles: 2OEC).**

## 4 Discussion

The 3OEC improves benthic flux measurements through the addition of the third concentration sensor, which eliminates errors that can be produced by time shifts between concentration and flow measurements. The averaging of the three instantaneous concentration signals also reduces signal variance and uncertainties that can arise from the disagreement in 255 two of three concentration sensor readings (Table 1). The measuring approach of the aquatic eddy covariance technique - determining fluxes at a distance from their origin - inherently produces data with relatively large variance. This can raise questions regarding their reliability. The fluxes recorded with the 3OEC are validated by the general agreement of the magnitudes and trends of the 3OEC, 2OEC and benthic advection chamber based fluxes (Huettel et al., 2020) all measured at the same study site, as well as benthic fluxes reported by Long (2021). Long's study site with carbonate sands at 6 m water 260 depth off Key Largo (Florida) was close (68.5 km distance) to ours, and the fluxes he recorded in June 2018 reached 5 mmol

$O_2$ m$^{-2}$ h$^{-1}$ during daytime and -3 mmol $O_2$ m$^{-2}$ h$^{-1}$ during nighttime, similar to the fluxes we measured (5.2, -2.8 mmol $O_2$ m$^{-2}$ h$^{-1}$).

The 3OEC improves benthic flux measurements in dynamic shelf environments. Here waves can produce artefacts in eddy covariance fluxes measurements (Berg et al., 2015), which in our 3OEC measurements were reduced by elimination of errors that can be caused by the spatial separation of concentration and flow measurements. Long (2021) proposed positioning the eddy covariance measurement point higher in the water column to decrease flux bias caused by waves. Our 35 cm measuring height was identical to that Long (2021) used for his measurements over Florida Keys sands and may have contributed to further reducing potential wave artifacts in our measurements. Our findings reveal that horizontal bottom currents dominated benthic flux modulation at our site despite the significant wave action (Figs. 2ab, 5a) in agreement with results of earlier studies that found an enhancing effect of current on flux in shallow shelf environments with permeable sediment (Berg et al., 2013; Chipman et al., 2016; Mcginnis et al., 2014). Continuous flow may be more effective than oscillating flow in driving advective pore water exchange in permeable sediments. In contrast to the steady pressure gradients that drive pore water exchange under continuous unidirectional bottom currents, wave orbital motion produces oscillating gradients, which enhance turbulence in the pore space of the sand (Horton and Pokrajac, 2009; Jouybari et al., 2020). This turbulence and inertial losses associated with the acceleration and deceleration of the pore flows may lessen the effectiveness of the oscillating pressure gradients for driving pore flows and interfacial water exchange.

Reduced uncertainties and higher precision achieved with the 3OEC facilitates more detailed analyses at higher temporal resolution (Fig. 6). This can produce new insights in the processes controlling fluxes at the seafloor. Co-spectra time series,

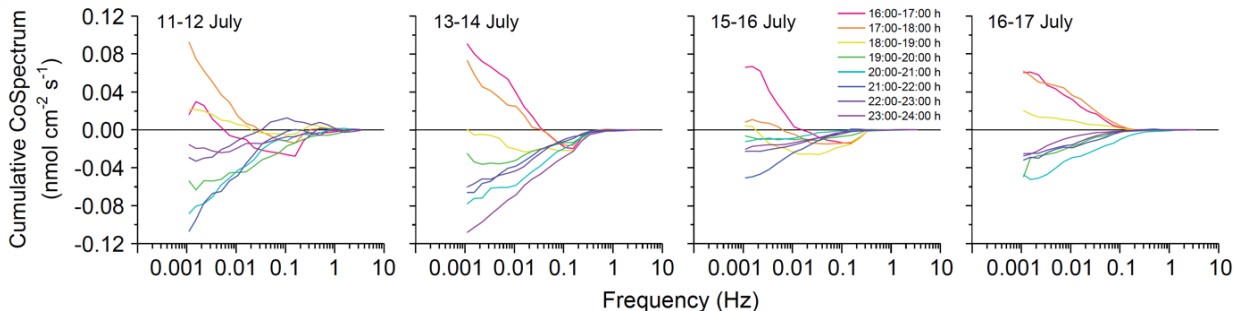

**Figure 6. Change of the cumulative co-spectra for the 3S $O_2$ flux during the deployment week. Cumulative co-spectra were calculated for hourly intervals from 16:00 to 24:00 for the four deployment periods (0.12 nmol cm$^{-2}$ s$^{-1}$ corresponds to 4.3 mmol m$^{-2}$ h$^{-1}$). Colours indicate the time periods for which the co-spectra were calculated, with red, orange and yellow (warm colours) depicting light periods before sunset (green). Blue and purple colours depict dark periods after sunset.**

plotted for hourly intervals from 16:00 to 24:00 for our 4 deployments indicate that during the transition from light to dark (Fig. 6, 16:00-19:00, warm colours), turbulence with a frequency < 0.1 Hz (larger eddies) still contained an upward directed

positive flux signal, while the higher frequency turbulence (smaller eddies) already carried a downward directed negative

flux signal. As the microphytobenthos photosynthetic $O_2$ production declined with the decreasing light intensity at the seafloor, flux switched from benthic $O_2$-release to $O_2$-uptake. The co-spectra suggest that the ensuing negative benthic flux signal initially was transported by the faster smaller eddies generated at the rough and $O_2$-consuming sediment-water interface, while the slower large eddies higher in the water column still carried the positive flux signal. The co-spectra document the gradual mixing of the smaller eddies with negative flux signal into the large eddies with positive flux signal,

i.e., the negative flux dip in the daytime co-spectra broadened with decreasing light conditions, expanding from the higher to lower frequencies. This eddy memory effect decreased with the general decrease of bottom current velocity during our field campaign, as less high frequency, small eddy turbulence is created at the sediment water interface at lower flow velocities (Lee and Cheung, 1999; Sleath, 1974). Consequently, the negative dip in the daytime co-spectra disappeared and the co-spectra appeared almost undisturbed in the last deployment (16-17 July) when bottom currents were low.


The $O_2$ fluxes recorded by the 3OEC characterized the coarse carbonate reef sands as sites of intense benthic production and coupled respiration. The nighttime $O_2$ consumption rates of the coral sand rival respiration rates measured in shallow shelf sediments with much higher organic carbon content (Glud, 2008; Middelburg et al., 2005; Hopkinson and Smith, 2005; Laursen and Seitzinger, 2002), and are within the range reported from other coral reef sands (Cyronak et al., 2013; Eyre et

al., 2013; Grenz et al., 2003; Rasheed et al., 2004; Wild et al., 2005; Wild et al., 2004). Since the coral sands at our site are low in organic carbon (< 0.3% dw) and occur in an oligotrophic subtropical reef environment with low water column chlorophyll and dissolved organic carbon content ($NO_3+NO_2$ < 0.2 µmol $l^{-1}$, $NH_4$ < 0.5 µmol $l^{-1}$, $PO_4$ < 0.05 µmol $l^{-1}$, Chl. a < 0.2 µg $l^{-1}$, DOC < 200 µmol $l^{-1}$, Huettel unpublished), a substantial sedimentary source of reduced compounds is required to maintain the observed high respiration rates. Our measurements point to benthic primary production as this source. The

compensation light intensity (intensity at which $O_2$ production exceeds respiration), reached at ~12 µmol photons $m^{-2}$ $s^{-1}$, and the high light utilization efficiency of 0.09-0.10 $O_2$ photon$^{-1}$ near the theoretical limit (0.12 $O_2$ photon$^{-1}$, (Brodersen et al., 2014; Attard and Glud, 2020)) indicated that the microphytobenthos could maintain excess production under cloudy conditions, identifying the sedimentary microalgae as source for the intense organic matter production and export. The estimated maximum production of ~10 mmol $m^{-2}$ $h^{-1}$ in the Florida carbonate sands (Fig. 5b), is in line with rates reported

for reef lagoon sediments in Moorea ($P_{max}$ 6.8 ± 0.5 mmol $m^{-2}$ $h^{-1}$ (Boucher et al., 1998)), New Caledonia ($P_{max}$ ~10 mmol $m^{-2}$ $h^{-1}$ (Clavier and Garrigue, 1999)) and the Great Barrier Reef ($P_{max}$ ~11 mmol $m^{-2}$ $h^{-1}$ (Eyre et al., 2013)). To put these rates into perspective, eddy covariance flux measurements over dense Mediterranean *Posidonia* seagrass meadows (13 m depth, PAR 300-400 µmol photons $m^{-2}$ $s^{-1}$) revealed daytime $O_2$ fluxes of 6.8 ± 0.7 mmol $m^{-2}$ $h^{-1}$ and nighttime fluxes -3.6 ± 0.4 mmol $m^{-2}$ $h^{-1}$ (Koopmans et al., 2020), i.e. rates of the same magnitude as measured in the microphytobenthos

communities. This suggests that the benthic metabolic activity in these shallow oligotrophic environments is largely controlled by light. The trends of nighttime respiration that mirrored those of daytime production (Figs. 2ab, 5a) indicate that at our site microphytobenthos drove the high $O_2$ consumption rates through its respiration and by producing highly

degradable organic matter that was promptly recycled by the benthic heterotrophic community. Factors contributing to the high microbial activity in the carbonate sands include the high specific surface area of the biogenic grains, their permeability to water and gases, the organic content of the grains, their chemical buffering capacity, and their light guiding characteristics (Marcelino et al., 2013; Huettel et al., 2014; Wild et al., 2006; Wild et al., 2005).

## 5 Conclusions

The deployments of the 3OEC demonstrate that the new instrument can improve the precision and reliability of benthic flux measurements. 3OEC fluxes in general were smaller, less variable and had smaller error margins than those produced by the conventional 2OEC eddy covariance instrument that was deployed next to the 3OEC. The advantages of the 3OEC may be most valuable in shallow energetic environments as reflected in the nighttime fluxes recorded by the 3OEC that differed significantly from those measured by the 2OEC. We believe that especially in dynamic settings, the improvements in flux determinations clearly outweigh the downsides associated with the slightly higher complexity of the 3OEC relative to conventional eddy covariance instruments with one or two solute sensors. As summarized in Table 1, the increases in setup time and costs are modest and may be justified by the improvement of quality and reliability of the flux data that can be achieved with the new instrument (Table 1).

**Table1. Comparison of 3OEC and 2OEC key characteristics. Approximate costs as of August 2021. Not included are costs forsupport frame (custom made) and additional sensors (e.g., light, temperature, same for both instruments)**

| Factors affecting instrument choice | 3OEC | 2OEC |
|---|---|---|
| **Performance during test deployments** | | |
| **Differences in observed magnitude of $O_2$ fluxes** | | |
| Normalized daytime flux relative to 3OEC | 100% | 106% ± 6% |
| Normalized nighttime flux relative to 3OEC | 100% | 151% ± 46% |
| **Precision during test deployments** | | |
| Normalized average standard deviation in daytime flux | 18% ± 9% | 50% ± 35% |
| Normalized average standard deviation in nighttime flux | 33% ± 11% | 51% ± 21% |
| **Time required for setup** | | |
| Instrument assembly from transport boxes to deployment-ready | 1-2 h | 1-1.5 h |
| Instrument deployment programming | 1 h | 0.5 h |
| Calibration | 1 h | 0.5 h |
| Total time for setup | 3-4 h | 2-2.5 h |
| Time for memory card removal and data download | 0.5 h | 0.5 h |
| Turn-around time for re-deployment | 1 h | 0.5 h |
| **Dimensions** | | |
| Height | 150 cm | 150 cm |
| Width | 120 cm | 120 cm |
| Weight above water | 40 kg | 30 kg |
| **Approximate costs** | | |
| 1xVector (Nortek) | 13279 $US | |
| 4xBatteries with 1xhousing (Nortek) | 1905 $US | |
| 1xdata logger (DataQ) | 695 $US | |
| 3x (2x 2OEC) $O_2$-meters with underwater fibre connectors (Pyroscience) | 6498 $US | 4332 $US |
| 3x (2x 2OEC) Ultrahighspeed $O_2$-optodes (Pyroscience) | 1392 $US | 928 $US |
| 1xElectronics under water housing for $O_2$-meters, 300 m (AGO Env.) | 2600 $US | |
| **Total** | **25969 $US** | **23339 $US** |

$O_2$ flux is a key indicator for changes in benthic metabolism and ecosystem health (Glud, 2008), emphasizing the need for reliable flux estimates. The aquatic eddy covariance technique arguably is the best available method for measuring flux at the seafloor as it does not alter activities of benthic fauna and flora, integrates effects of patchiness, and accounts for the effects flow, light, temperature as well as the supply of electron donors and acceptors that affect the fluxes. The increased precision and reliability of the 3OEC data allow improved modelling and ecological interpretation. In many environmental measuring tasks, a basic, inexpensive instrument can produce data that are relatively close to the "true" values, and the effort and cost for improving the quality of these data typically increase exponentially with gain in data accuracy and precision. The recent developments in affordable optode technology allow setting up a 3-optode aquatic eddy covariance instrument at modest extra cost relative to a conventional 2-optode instrument. The 3OEC is an improved eddy covariance instrument that requires less data postprocessing and produces flux data of higher quality and reliability. It presents a hardware solution that permits flux measurements also in dynamic shallow shelf environments and is an unmatched instrument to study and improve flux extraction methodologies. We consider the $O_2$ fluxes produced by this instrument for the permeable reef sands as some of the most realistic flux estimates achievable with present day technology.

## 6 Appendix A

**Table A1: Specifications of the Pyroscience™ OXR430-UHS retractable oxygen minisensors used in this study**

| | |
|---|---|
| Optical $O_2$ fibre sensor type | Pyroscience™ OXR430-UHS |
| Fibber diameter | 430 µm |
| Optimal measuring range | 0-720 µmol $l^{-1}$ |
| Maximum measuring range | 0 - 1440 µmol $l^{-1}$ |
| Detection limit | 0.3 µmol $l^{-1}$ |
| Resolution at 1% $O_2$ | 0.16 µmol $l^{-1}$ |
| Resolution at 20% $O_2$ | 0.78 µmol $l^{-1}$ |
| Accuracy at 1% $O_2$ | ± 0.31 µmol $l^{-1}$ |
| Accuracy at 20% $O_2$ | ± 3.13 µmol $l^{-1}$ |
| Temperature range | 0 - 50°C |

355 **Table A2: Specifications of the Pyroscience™ FireStingO$_2$-Mini oxygen meter used in this study**

| | |
|---|---|
| Pyroscience™ FireStingO$_2$-Mini | Single sensor module, |
| Oxygen port | 1 fibre-optic ST-connector |
| Temperature port | 4-wire PT100, -30°C-150°C, 0.02°C resolution, ±0.5°C accuracy |
| Dimensions and Weight | 67 x 25 x 25 mm, 70 g |
| Measuring principle | Luminescence lifetime detection (REDFLASH) |
| Excitation Wavelength | 620 nm (orange-red) |
| Emission wavelength | 760 nm (NIR) |
| Maximum sampling rate | 20 Hz |
| Interface | Serial interface (UART), ASCII communication protocol |
| Analog output | 0 - 2.5 V DC, 14-bit resolution |
| Power requirements | Max. 70 mA at 5 V DC from USB (typ. 50 mA) |

**Table A3: Specifications of the NORTEK Vector acoustic Doppler velocimeter used in this study**

| Sensor | Range | Accuracy | Precision/Resolution |
|---|---|---|---|
| Velocity | ±0.01, 0.1, 0.3, 1, 2, 4, 7 m $s^{-1}$ | ± 0.5% | ± 1% |
| Pressure | 0-20 m (shallow water version) | 0.5% (full scale) | < 0.005% of full scale |
| Temperature | -4 to +40 °C | 0.1 °C | 0.01 °C |
| Compass | 360º | 2º | 0.1º |
| Tilt | < 30° | 0.2° | 0.1° |

## 7 Data availability

Current flow, pressure, and oxygen concentrations recorded by the 3OEC-instrument (doi:10.26008/1912/bco-dmo.849934.1), reference temperature and dissolved oxygen (doi:10.26008/1912/bco-dmo.849915.1) and PAR data (doi:10.26008/1912/bco-dmo.849979.1) recorded 11-17 July 2017 are available at the Biological and Chemical Oceanography Data Management Office (BCO-DMO, https://www.bco-dmo.org/).

## 8 Author contributions

AM deployed the 3OEC, analysed the data, and wrote the first version of the manuscript. MH designed and built the 3OEC instrument. MH and PB contributed to the data analysis and the preparation of the manuscript.

### 9 Competing interest statement

The authors declare that they have no conflict of interest.

## 10 Acknowledgments

We thank the staff of the FIO Florida Keys Marine Laboratory for help with instrument deployments and sample collection. The research was conducted under NOAA permit FKNMS-2012-137-A2 and was supported by NSF grants OCE-1334117 OCE-1851290, and OCE-1061364.

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
