# Peer review of "Technical Note: Novel triple O2 sensor aquatic eddy covariance instrument with improved time shift correction reveals central role of microphytobenthos for carbon cycling in coral reef sands"

_Biogeosciences, 2021_

## Author Comment (AC1)

**Response to the comments and questions of Dirk de Beer (Reviewer#1)**

We thank Dirk de Beer for the review that helped improving this manuscript and Jack Middelburg for handling our paper. We addressed all reviewer questions and comments, and our responses and actions are listed point by point below.

**Reviewer#1**

This is an interesting and well written method paper. Although a method paper, some observations are thought-provoking. The ecological aspects may deserve a dedicated paper rather than 'hiding' the data in a technical note.

> Response: We thank this reviewer for the positive comment. We decided to keep some of the ecological aspects in this technical note as we believe that demonstrating that this instrument can produce "thought-provoking" data may convince scientists to adopt this technique.

Important is if the more complex method is better, and worth the extra costs and effort. In the end the results with 3 sensors are similar those from the 2 sensor design, so would the conclusion be that 1 is enough? Line 164 states that the differences were not statistically significant. The advantages of the 3 sensor system is at the moment not well explained. This is my main issue with this manuscript.

> Response (P15L320): The reviewer addresses a key point that was also raised by Reviewer#2. We took up the suggestion of Reviewer#2 and re-analyzed the data using a paired t-test. The new analyses revealed that the nighttime fluxes measured by the 3OEC over the study differed significantly from those measured by the 2OEC, while the daytime fluxes did not. We now summarize the advantages of the 3OEC in the conclusions section and added a Table that compares performance and costs the 3OEC and 2OEC. We added the following paragraph to the conclusions: "The deployments of the 3OEC demonstrate that the new instrument can improve the precision and reliability of benthic flux measurements. 3OEC fluxes in general were smaller, less variable and had smaller error margins than those produced by the conventional 2OEC eddy covariance instrument that was deployed next to the 3OEC. The advantages of the 3OEC may be most valuable in shallow energetic environments as reflected in the nighttime fluxes recorded by the 3OEC that differed significantly from those measured by the 2OEC. We believe that especially in dynamic settings, the improvements in flux determinations clearly outweigh the downsides associated with the slightly higher complexity of the 3OEC relative to conventional eddy covariance instruments with one or two solute sensors. As summarized in Table 1, the increases in setup time and costs are modest and may be justified by the improvement of quality and reliability of the flux data that can be achieved with the new instrument (Table 1)."

The introduction is complete, but the focus could be a bit shifted. It should start with the method rather then the carbonate sands in reefs, as the method is more generally applicable than in reef sands. The strength of the eddy covariance method is that it can measure exchange fluxes with all benthic ecosystems, with few exceptions, and from those fluxes infer metabolic activities.

> Response (P1L29): We followed the suggestion of the reviewer and changed the sequence in the introduction, addressing the new method first before introducing the reef carbonate sand environment used as testing site. The abstract was changed accordingly.

As the paper describes an improved method, the current issues should be better introduced. At the moment designs with 1 and 2 sensors are used. The reasoning for using a 3th sensor should be better explained. My understanding is that the variations in flow direction make it difficult to link the local flow direction and speed to the local oxygen concentration, as the flow and concentrations are measured a few cm apart. The corrections are now made by continuously time-shifting the oxygen signal by the measured flow speed and direction, and thus synchronize both. This is apparently not perfect. Exactly how 3 sensors can improve things should be described in detail, best with sketches.

> Response (P2L43): We re-wrote the respective section and added a figure (Fig. 1e): "Since the fluxes are calculated from minute concentration changes measured at a high frequency required to account for all water movements that transport the solute, it is critical that the flow data are accurately aligned in time with the associated solute data. This presently is a potential source of error as conventional aquatic eddy covariance instruments cannot measure current velocities and solute concentrations in the same location. Typically, current velocities are recorded with an acoustic Doppler velocimeter (ADV), and solutes with a fast-responding electrochemical or optical sensor (Kuwae et al., 2006; Berg et al., 2003; Reimers et al., 2012; Attard et al., 2016; Donis et al., 2016; Glud et al., 2010; Mcginnis et al., 2014; Lorke et al., 2013; Huettel et al., 2020).

The tip of the solute sensor in these instruments is positioned at a few centimetres horizontal distance from the ADV's measuring volume to prevent disturbances of the flow and the acoustic ADV signal. This spatial separation of flow and solute measurements causes a misalignment between the two measurement time series, which requires a time shift correction of the data. In environments with dynamic currents, this misalignment changes continuously as direction and velocity of the turbulent flow varies. Algorithms were developed that shift the $O_2$ data in time such that they are synchronized with the velocity data (Mcginnis et al., 2008a; Berg et al., 2015; Reimers et al., 2016). A common procedure is to move a short sequence (e.g., 15 min) of solute data in time relative to the current flow data recorded at that time until a maximum in flux is reached. In steady unidirectional flow, this procedure largely can eliminate time shift errors, but it is difficult to apply an effective correction in dynamic settings (Donis et al., 2015; Reimers et al., 2016). Since the rapid changes in solute concentration and vertical flow velocity are relatively small and affected by signal noise, a distinct maximum in flux may not be found when time shifting the data, which can result in erroneous corrections and fluxes. Furthermore, wave orbital motion in shelf environments produces oscillating bottom currents that may change in magnitude and direction at the time scale of seconds, complicating a correct alignment of the data and producing further potential sources of uncertainty in the flux calculations. In the conventional eddy covariance instruments with one or two solute sensors, the cumulative effect of small errors in the time shift correction thus can lead to significant under- or overestimates of the flux, which in extreme cases can reverse the direction of the calculated flux relative to the true flux (Berg et al., 2015; Reimers et al., 2016). To remove this potential source of error, we designed a triple $O_2$ sensor eddy covariance instrument (3OEC) that eliminates the uncertainties caused by the spatial separation of flow and concentration measurements."

The habitat in which the tests are done seems not the most challenging, as the flow direction is rather constant. Not much wave action occurs at 10 m depth.

Response (P3L68): It is correct that more challenging habitats may exist (e.g. wave braking zones, environments with large structures on the seabed, kelp forests) but in such environments basic assumptions on which eddy covariance flux measurements are based (i.e. that a bottom current with a steady state mean flow and oxygen concentration reaches the instrument unobstructed after passing a measuring area with relatively constant surface roughness (Massman and Lee, 2002;Baldocchi, 2003;Kuwae et al., 2006;Berg et al., 2007) are violated such that corrections are not possible. Considering these basic assumptions, our testing site with clear water and substantial wave action was challenging for eddy covariance measurements as explained in the text: "The instrument was tested in the Florida Keys at an exposed inner shelf site with carbonate sands, clear oligotrophic water and substantial wave action, i.e., in an environment considered challenging for eddy covariance measurements due to the low particle concentrations in the water (ADV measurements rely on sound reflection from particles) and the dynamic flows (causing the data misalignments addressed in this study)." and (P11L236) "At our study site, waves were relatively high for this shallow environment (wave height up to 10% of water depth), and wave orbital motion influenced water movement and pressure near the seafloor during the entire study (e.g., Fig. 3e).". The relatively small orbital motion in the boundary layer at our study site may be more challenging for eddy covariance measurements than large orbital motions with slower changes in flow direction under large waves.

The method deserves better explanation. Fig. 1 is good in the sense that is complete, but one needs a microscope. Crucial parts need to be magnified: the yellow cylinder and red triangle in 1b (in my high quality printout an amorphous colored little blob), the yellow ring in 1c is invisible, 1d can be omitted as it has no information. Is the pink stuff in 1a an error or something real?

Response (Fig. 1): The reviewer has a good point, and we enlarged the images and replaced Fig. 1d with a heat map visualizing the equivalence of the average of the three sensor readings and the concentration at the centre of the ADV volume. The "pink stuff" is flexible pink plastic tape that was attached to the frame to allow divers assessing the flow direction.

A mathematical proof is needed for the crucial assumption that the average of the 3 sensors is the concentration in the geometric center of the sensors. This is crucial for the paper and should be provided. From Fig 8 one can see that a lot of detailed post processing is needed to get to the average!

Response (Figure 1e): We followed the request of the reviewer and added a mathematical proof that the average of the 3 sensors is the concentration in the geometric center of the sensors, and a schematic visualizing that proof (Fig. 1e). The legend of Fig. 1e explains the approach and reads: "Proof of the equivalence of the average of the three sensor readings and the concentration at the centre of the equilateral triangle defined by the positions of the three optode sensor tips (red triangle in (b) and (c)). We postulate that the concentration gradients between the sensor tips are linear (see also text). Accordingly, the half-way point of a side of the triangle corresponds to the average concentration measured by the optodes at the two endpoints of that side. Applying the law of sines and triangle congruence criteria (transitive property of congruence, angle bisector theorem, converse of angle bisector theorem), the concentration at the centre point (M) of the equilateral triangle

(ABC) equals the concentration at the vertex (M) of the right triangle defined by one vertex of the equilateral triangle (C in above example) and the midpoint ((A+C)/2 in above example) between that vertex and the second vertex (A in above example) on that line. According to the congruence criteria, this is valid for analogous, congruent right triangles constructed on the other sides of the triangle. As these triangles are based on the average concentration of two vertices of the equilateral triangle, it follows that the centrepoint concentration is equivalent to the average concentrations measured by the three sensors. The heat map visualizes the equivalence of the concentrations at the centre of the equilateral triangle calculated using this approach and the average of the three sensor signals."

Further explanations are given in the text (P3L88): "The new 3OEC instrument utilizes a data averaging approach to remove misalignments in time caused by the spatial separation of flow and $O_2$ measurements and thereby eliminates potential errors caused by time shift corrections. The 3OEC measures simultaneously with three $O_2$ fibre optodes positioned at 120 degrees angular spacing in the same horizontal plane around the centre point of the water volume where current flow is measured by the ADV (Fig. 1). Assuming approximately linear concentration gradients within the 6.4 cm distance between the $O_2$ sensors - justifiable according to planar optode readings of water column oxygen distributions in turbulent flows (Glud et al., 2001; Larsen et al., 2011; Oguri et al., 2007) - the $O_2$ concentration at the location where the flow is measured can be calculated through averaging of the three simultaneous sensor signals. The three optode tips, positioned at the corners of an equilateral triangle, present three equidistant points on a circle with the measuring volume of the ADV at its centre. For linear concentration gradients between the three measuring points, it can be proven with the law of sines that the $O_2$ concentration at the centre of this circle corresponds to the average of the three sensor signals (Fig. 1e)."

Comparing Fig 2 and 4, there seems to be no relation between ecosystem activity and hydraulics. There seems a relation between the integral of the irradiance (photons/day) and both productivity at day and the respiration at night. Would be good to do some calculations on this. Would be indeed logical and understandable if the metabolic activity, including respiration at night, is controlled by the light input.

Response (P11L237): We here respectfully partly disagree with the reviewer. Figures 2ab and 5a depict the decreases in flux with decreasing bottom current velocity. Although the number of data points is limited, this result is supported by both independently measuring eddy covariance systems and reveals the influence of bottom current strength on the flux. As mentioned in the text, the light conditions on the different measuring days were similar, excluding light as the cause for the observed flux decrease over the study period. We therefore concluded that the decrease in flow produced the decrease in flux over the study period, which is supported by eddy covariance measurements over other permeable sand beds (e.g. Berg et al., 2013; Chipman et al., 2016; Mcginnis et al., 2014). That said, we in agree with the reviewer that productivity at day and the respiration at night is related to the integral of irradiance.

We agree with the reviewer that the benthic metabolic activity in general is largely controlled by light input, which we phrased in the text (P14L311): "To put these rates into perspective, eddy covariance flux measurements over dense Mediterranean *Posidonia* seagrass meadows (13 m depth, PAR 300-400 μmol photons $m^{-2}$ $s^{-1}$) revealed daytime $O_2$ fluxes of $6.8 \pm 0.7$ mmol $m^{-2}$ $h^{-1}$ and nighttime fluxes $-3.6 \pm 0.4$ mmol $m^{-2}$ $h^{-1}$ (Koopmans et al., 2020), i.e. rates of the same magnitude as measured in the microphytobenthos communities. This suggests that benthic metabolic activity in these shallow oligotrophic environments is largely controlled by light. The trends of nighttime respiration that mirrored those of daytime production (Figs. 2ab, 5a) indicate that at our site microphytobenthos drove the high $O_2$ consumption rates through its respiration and by producing highly degradable organic matter that was promptly recycled by the benthic heterotrophic community."

The cited numbers in L 211 and 212 are 1000x off. Actually Koopmans measured similar (to this study) metabolic rates in an equally illuminated, also oligotrophic, ecosystem. That the two ecosystems, although totally different (seagrass canopies versus reef MBP), are similarly productive at the same light level seems meaningful. Both benthic photosynthesis and respiration are controlled by light input.

Response (P14L313): We thank the reviewer for catching these typos, the magnitudes were corrected.

The oxygen exchange rates measured in this study are well in the reported ranges. These ranges are very wide. L 195 states 'matching or exceeding', but rates are truly about average, sometimes below, sometime above average (see Fig 6).

Response (P14L296): We corrected our statement that now reads "The nighttime $O_2$ consumption rates of the coral sand rival respiration rates measured in shallow shelf sediments with much higher organic carbon content (Glud, 2008; Middelburg et al., 2005; Hopkinson and Smith, 2005; Laursen and Seitzinger, 2002), and are within the range reported from other coral reef sands (Cyronak et al., 2013; Eyre et al., 2013; Grenz et al., 2003; Rasheed et al., 2004; Wild et al., 2005; Wild et al., 2004)."

It is interesting that small eddies can transport oxygen in a different direction as big
eddies. These go thus in different directions. Can a physical interpretation be given for this
phenomenon, and an indication of their sizes? The big eddies persist for several hours, is
that correctly understood? This phenomenon, if true, requires a larger discussion, as it
may have a large impact on the interpretation of such data.

> Response (P13L279): We agree with the reviewer that this is an interesting finding that needs further exploration and discussion. This would go beyond the scope of this manuscript. We presently are preparing a paper that analyzes this observation in detail.

Some minor points:
The ecosystem is net autotrophic, and since no organic deposits are build up, the
produced organics must be exported. Any suggestions how?

> Response (Fig. 5b): This is an interesting question and our research did not identify one process that would remove the organic matter. Likely processes that remove organic matter from the area are episodic storm events that cause major sediment resuspension (tropical depressions, typical for late summer) and abundant grazers that move in and out the area with the tide (demersal fish). It is also possible that the ecosystem cycles through autotrophic and heterotrophic periods, balancing summertime organic matter build-up and winter time consumption of that material. We don't have data to support theses speculations.

'elevation' in legend Fig. 8 must be defined and explained.

> Response (P10L208): We added the following explanation: "Elevation $z$ expresses the instantaneous relative elevation of a water parcel that is moved up and down at the velocity $V_z$ and can be estimated as $z = \int V_z\, dt$ (Berg et al., 2015)"

---

## Author Comment (AC2)

**Response to the comments and questions of Conrad Pilditch (Reviewer#2)**

We thank Conrad Pilditch for the review that helped improving this manuscript. We addressed all questions and comments, and our responses and actions are listed point by point below.

Overall this is a very interesting paper and one I enjoyed reading very much. It tests whether adding additional O2 sensors to an eddy-covariance instrument improves aliasing in data due to the separation of velocity and O2 sampling locations. Given the increasing use of eddy covariance measurements to estimate benthic primary production/respiration at scale technological improvements are timely and welcome. I am very supportive of this paper however there are a number of elements that if considered in revision could improve the focus and clarity of the manuscript.

> Response: We appreciate the positive comments of the reviewer and that the manuscript was perceived as timely.

My main comment addresses the multiple elements to this paper. It seems to bounce around between a confirmation of the fact that shallow water permeable carbonate sands are hot spots of benthic primary production and organic matter processing and testing whether additional O2 sensors improves the precision of flux measurement. Given the paper is submitted as a technical note it could be improved (and shortened) by focusing on the increase in performance of adding additional sensors. The sections of the paper discussing the high production/respiration rates of carbonate sands confirms previous studies (see Fig 6) and in my opinion distracts from the method which is generalizable to many systems.

> Response: We appreciate this comment that is similar to the one phrased by Reviewer#1, and in the revised manuscript we now shifted the focus more onto the method and removed some of the ecological interpretations. These changes affect the abstract, introduction and discussion sections. Sections addressing carbonate sand were shortened and placed after the sections addressing the technology. The changes are also described in the responses to Reviewer#1.

If using eddy-covariance techniques I would want to know exactly what gains could be made by adding sensors as the additionally increases costs and/or may reduce the ability to spatial replicate units. These trade-offs are import – is it more important to increase the precision at one location or potentially increase the number of locations at which flux measurements are made to assess spatial variability? So the questions I would like answered explicitly are what improvements are made my adding sensors in terms of precision and do these improvements vary with hydrodynamic setting (eg. uniform steady flow vs more wave dominated flows), do these improvements really matter in system with high natural variability in fluxes and what other conditions/settings need testing to confirm the value of sensor additions. Revising the manuscript (mainly editing the Introduction/Discussion) with this comment in mind I think would result in a much more assessable paper with a tighter focus. In short make the technical note more about the method than the system in which it was tested.

> Response: (P12L252). We added a paragraph and table that summarizes the characteristics of the new 3OEC and the 2OEC and highlights the advantages of the new instrument. The table documents that adding sensors increases the cost of the instrument relatively little (~10%) and the effort for setting up the new instrument is very similar to the effort needed for a conventional instrument. The discussion now is more focused on the new method.

Specific Comments

Ln 160 Please provide more detail on what data was used in t-test comparing the 3OS and 2OS. The DF indicates 7 data points – were these the average of the 15 min blocks across the four sample dates? I am assuming that both the 3OEC and 2OES systems were synched so perhaps a better test may have been a paired t-test where you ask whether the difference between the data is $<>$ than 0.

> Response (P8L192). We followed the suggestion by the reviewer and now apply a paired t-test for comparing the 3OEC and 2OEC fluxes. According to this test, the nighttime fluxes between the two instruments were significantly different, while the daytime fluxes were not. We changed the text accordingly which now reads (P7L191): "Average 3OEC daytime fluxes were 7% lower and nighttime fluxes 38% lower than the respective 2OEC fluxes (day: 5.6 ± 0.8(SE) night: -3.9 ± 0.5(SE) mmol

m$^{-2}$ h$^{-1}$). The difference in the nighttime fluxes between the two instruments was statistically significant (p= 0.04685, p(x≤T) = 0.02342, T =-3.268, DF=3), while the difference in daytime fluxes was not (p=0.08077, p(x≤T) = 0.9596, T = 2.5944, DF=3).”

Line 165 Two sentence paragraph that does not make sense on its own

Response (P11L237). Thank you for pointing this out. The paragraph was rewritten and now reads: “Yet, fluxes scaled with the average unidirectional bottom current velocity, which slowed during the deployment week (~30 mmol m-2 h-1 flux increase or decrease per m s-1 flow decrease, Fig. 5a), and not with significant wave height (R2<0.04, Fig. 2ab) that increased during the study except the last day.”.

Fig 3 – Add the p values for the regression statistics and clarify what data is being average for each of the visible data points. If the data represents averages of 15 min intervals then surely there is a variation in mean current velocity between intervals that should be plotted as an error term?

Response (Figure. 5). We added the standard errors for the average velocities and included the following explanation in the legend “The data points indicate the average fluxes calculated for light daytime and dark nighttime periods, separated at 20:00, plotted against the average flow velocity for the respective time periods. The compromised data point from the 16 July 2OEC deployment was excluded from the regression (grey circle). Error bars represent standard error.”. The four data points (2OEC night 3 data points) available for the regressions do not allow meaningful regression statistics and we removed the R$^2$ values. The combined nighttime data of the two instruments produce a statistically significant trend, the daytime data don’t. Nevertheless, the consistent and almost identical trends (with respect to slope) observed in the two independently measuring instruments as well as agreeing reports in the literature documenting flux enhancement by flow in permeable sediment, e.g. (Berg et al., 2013;Chipman et al., 2016;McGinnis et al., 2014) support our interpretation that fluxes increased with flow velocity.

I would also like to see what if any difference results from the generated PI curves from the 2OEC and 3OEC systems – are the fits better (r2) are the fitted parameters known with greater precision and does this matter?

Response (P11L244). The 3OEC improves the precision which improves the reliability of the PI curves, however, due to the scatter in the data, the differences between the predicted maximum gross benthic primary production rates were statistically not significantly different. The revised text section now reads: “Improved precision and the generally lower fluxes in the 3OEC were reflected in the community photosynthesis-irradiance (PI) curves (Bernardi et al., 2015). The 3OEC predicted a slightly lower maximum gross benthic primary production (GPP) of 9.9 mmol O$_2$ m$^{-2}$ h$^{-1}$ (R$^2$: 0.999) than the 2OEC (10.7 mmol O$_2$ m$^{-2}$ h$^{-1}$, R$^2$: 0.998, Fig. 5b) as well as a lower light utilization efficiency (LUE, ratio between GPP and PAR, 3OEC LUE 12.3% lower than 2OEC LUE at 10 µmol photon m$^{-2}$ s$^{-1}$ and 7.4% lower at 350 µmol photon m$^{-2}$ s$^{-1}$ (Fig. 5c)). Due to the scatter in the data, these differences in GPP maxima and LUE were statistically not significant.”

Fig 5 Are there any corrections applied to data in the (a)? That is, is the variation observed between individual sensors a function of sensor performance vs data that has not been corrected for R, S, T & W.

Response (Figure 4). We mention in the legend “(all data in (a) and (b) are STW- corrected)”

Line 180 The order is a little illogical – when looking at Fig 5 c&d I see a T correction was included in the 3OEC data processing – yet in the Introduction it was emphasised that the advantage of using 3 sensors was to avoid needing to do this. It is not until the Discussion that this is discrepancy is explained. I would suggest that Fig 8 & text goes into the Results to explain/justify this correction. Alternatively improve the legend to Fig 5 and point to the Discussion for explanation.

Response (Fig. 3). We followed the suggestion of the reviewer and moved Fig. 8 to the results section (now Fig. 3).

As mentioned above the Discussion might be better focused on the comparisons between sensor configurations. To aid this the authors could consider adding a summary table that summarises the increased in precision and compares this the natural variability and provide some 'cost-benefit' analysis for investigators. Are there conditions where a 1 or 2 sensor system may give similar results to a 3 sensor system and where should researchers favour a 3 sensor system?

Response (P15L323). The summary table is a good idea and we added this table as suggested by the reviewer. In addition, we included the following paragraph with the table: "The deployments of the 3OEC demonstrate that the new instrument can improve the precision and reliability of benthic flux measurements. 3OEC fluxes in general were smaller, less variable and had smaller error margins than those produced by the conventional 2OEC eddy covariance instrument that was deployed next to the 3OEC. The advantages of the 3OEC may be most valuable in shallow energetic environments as reflected in the nighttime fluxes recorded by the 3OEC that differed significantly from those measured by the 2OEC. We believe that especially in dynamic settings, the improvements in flux determinations clearly outweigh the downsides associated with the slightly higher complexity of the 3OEC relative to conventional eddy covariance instruments with one or two solute sensors. As summarized in Table 1, the increases in setup time and costs are modest and may be justified by the improvement of quality and reliability of the flux data that can be achieved with the new instrument (Table 1)"